# GENERATIVE VISUAL INSTRUCTION TUNING

## ABSTRACT

We propose to use automatically generated instruction-following data to improve the zero-shot capabilities of a large multimodal model with additional support for generative and image editing tasks. We achieve this by curating a new multimodal instruction-following set using GPT-4V and existing datasets for image generation and editing. Using this instruction set and the existing LLaVA-Finetune instruction set for visual understanding tasks, we produce GenLLaVA, a Generative Large Language and Visual Assistant. GenLLaVA is built through a strategy that combines three types of large pretrained models through instruction finetuning: Mistral for language modeling, SigLIP for image-text matching, and StableDiffusion for text-to-image generation. Our model demonstrates visual understanding capabilities superior to LLaVA and additionally demonstrates competitive results with native multimodal models such as Unified-IO 2, paving the way for building advanced general-purpose visual assistants by effectively re-using existing multimodal models.

## 1 INTRODUCTION

The field of multimodal models has become increasingly popular in the research community as they are one of the key building blocks for general-purpose assistants (Achiam et al., 2023; Gemini Team et al., 2023; Bai et al., 2023). One of the main directions researchers have pursued is to combine Large Language Models (LLMs) with Vision Models for multimodal tasks *i.e.* creating LVLMs. The recently proposed LLaVA model (Liu et al., 2023b) is among the latest wave of works that have demonstrated the effectiveness of instruction tuning for multimodal models (Liu et al., 2024a; Zhao et al., 2023; Wang et al., 2023a; Karamcheti et al., 2024). In these works, a two-stage pipeline is followed: (1) multimodal pre-training where the unimodal models are combined and trained on a large corpus of captioning data Schuhmann et al. (2022); Ordonez et al. (2011); and (2) a supervised fine-tuning (SFT) Liu et al. (2024a;b) stage where the model is trained on domain-specific data and enables it to better perform various downstream tasks of interest.

Visual generation is another research direction that combines visual and language modalities. There are two common approaches for text-to-image generation. One approach employs diffusion models (Rombach et al., 2022), which have shown unprecedented performance in image synthesis, becoming the de facto method for visual generation. The other line of work converts visual content into discrete tokens using vector quantization (VQ) and then leverages an autoregressive transformer for high-quality image synthesis (Chang et al., 2022; Lee et al., 2022).

As visual understanding and generation capabilities advance rapidly and independently, a growing trend is to combine these into a unified Large Multimodal Model (LMM). There are two main approaches to achieving such unification. Many LMMs (Koh et al., 2024; Sun et al., 2024b; Fu et al., 2024) produce conditional embeddings to be used by a pretrained diffusion model for image generation. On the other hand, there are LMMs (Lu et al., 2024a; Chameleon Team, 2024; Yu et al., 2023) that adopt VQ encoders to project visual inputs into discrete tokens and use the same next-token prediction paradigm as Language Models.

There has been a considerable amount of work building on top of the LLaVA model ranging from image generation (Koh et al., 2024; Sun et al., 2024b;a), grounding (You et al., 2024), image editing (Fu et al., 2024) to video understanding (Lin et al., 2024). These works share the same principles; they extend the ideas of visual instruction tuning to one or more capabilities. However, after adding a new capability (i.e., image generation), the resulting models often lose some, if not

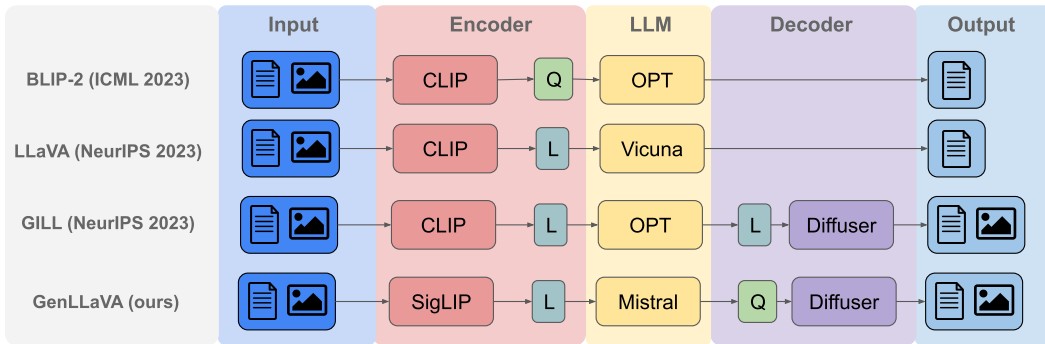

Figure 1: Comparison of GenLLaVA against recent architectures. Unlike BLIP-2 (Li et al., 2023), we use a Linear projector similar to the LlaVA architecture (Liu et al., 2023b). Generation capabilities are added using a diffusion model, but unlike GILL (Koh et al., 2024), we use a Q-former as the generation head. Finally, our model benefits from using a stronger visual encoder, namely SigLIP(Zhai et al., 2023); a stronger LLM, namely Mistral-7b (Jiang et al., 2023); and a stronger diffuser, namely SDv1.4 (Rombach et al., 2022). * L stands for Linear projection, and Q stands for Q-former resampler.

most, of their visual and language understanding capabilities. We propose a strategy that leads to a model that can perform generative tasks while retaining multimodal understanding capabilities.

The works more similar to our own are GILL Koh et al. (2024) and SEED-X Ge et al. (2024). Unlike GILL, which maps LLM embeddings to text embeddings for a pre-existing diffusion model, our method directly injects LLM-generated representations into the visual latent space of the diffusion process. This architectural difference enables GenLLaVA to achieve more nuanced image editing and generation capabilities. Compared to SEED-X, which employs task-specific checkpoints (SEED-X-I for image understanding, SEED-X-Gen for generation, and SEED-X-Edit for editing), GenLLaVA achieves a truly unified approach. While SEED-X provides a general-purpose checkpoint, its performance significantly degrades when compared to task-specific models. In contrast, GenLLaVA balances all modalities effectively within a single checkpoint, eliminating the need for task-specific models and offering seamless transitions between understanding, generation, and editing tasks without significant performance trade-offs.

In this paper, we present *generative visual instruction tuning*, an approach in which we teach a Large Multimodal Model (LMM) image understanding, image generation, and image editing tasks without diminishing the performance of each individual capability. (See Fig. 1 for an overview of our method.) To our knowledge, this is the first time such capability has been achieved, and our findings pave the way for building a general-purpose visual assistant. Our contributions are the following:

- *Generative multimodal instruction-following data.* Inspired by Liu et al. (2023b), which curated an instruction set for image understanding tasks, we curate a multimodal instruction tuning set that combines image understanding, image generation, and image editing data.

- *A single composite model*, *i.e.* GenLLaVA, which unifies visual understanding and generation using open-source models. GenLLaVA is trained using a single-stage training recipe, unlike its predecessor LLaVA Liu et al. (2024a).

- *Open source.* We will publicly release our generated multimodal instruction data, code to replicate our results, model checkpoints, and a visual chat demo.

## 2   RELATED WORK

**Large Multimodal Models (LMMs).** Large Multimodal Models (LMMs) refer to large language models that can understand various modalities beyond human language. Some research efforts are focused on combining image, audio, video, and other modalities with language  (Zhan et al., 2024; Lu et al., 2024a), while others aim to enhance the fusion of vision knowledge and language. For example, BLIP-2 (Li et al., 2023) created a large-scale image captioning dataset and paired a language model

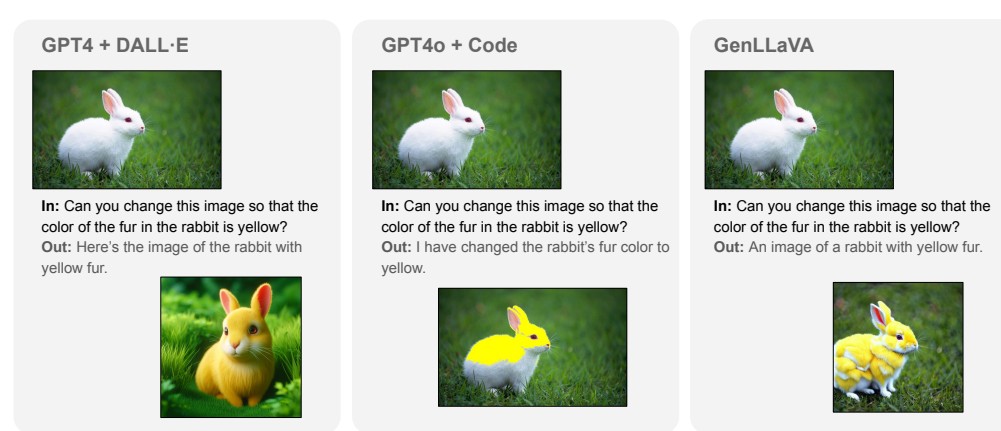

Figure 2: Editing capabilities of our model. GPT4 currently uses a version of the DALLE text-to-image model as a tool and, hence, is not directly able to edit images. GPT4o instead uses tools through Python-generated code to accomplish the requested action. Our model, GenLLaVA, connects input features obtained from CLIP to a language model that also produces output embeddings for a text-to-image StableDiffusion model, achieving an end-to-end editing task with a multimodal model.

with a vision encoder to produce a robust multimodal model. Following this, LLaVA (Liu et al., 2023b) developed a cost-effective approach to train an advanced LLM through visual instruction tuning. Although LLaVA-NeXT (Liu et al., 2024b) improved performance for single-image tasks, it required over 2,000 tokens per image, which is about four times more than the original LLaVA. More recent models such as QwenVL (Bai et al., 2023), CogVLM (Wang et al., 2023b), and Yi-VL (Young et al., 2024) follow architectures similar to those of LLaVA. Our proposed method not only focuses on models for multimodal understanding but also on adding generative capabilities to such models.

**Diffusion-based LMMs for visual generation** We review works that combine diffusion with autoregressive prediction to create LMMs for generative tasks. For instance, GILL (Koh et al., 2024) translates the hidden representations of an LLM into embeddings that correspond to a text-to-image model by learning a neural network to perform efficient mapping using the text encoder of the diffusion model. MGIE (Fu et al., 2024) adapts the text embedder, image input adapter, and LM head output parameters of an LMM jointly with a diffusion model for image editing from instructions. DreamLLM (Dong et al., 2024) uses the same paradigm as GILL and MGIE but instead trains on interleaves documents for visual generation and understanding synergy. Show-O (Xie et al., 2024) uses a bidirectional casual mask on the visual tokens combined with the next-token prediction objective but uses an extra masking loss similar to MaskGIT (Chang et al., 2022). in a single unified model to understand and generate both discrete and continuous modalities.

**Token-based LMMs for visual generation** We review works that project visual features into discrete tokens and use next-token prediction for generative tasks. AnyGPT (Zhan et al., 2024) discretizes data from multiple modalities, extends the existing LLM vocabulary to add the extra modalities, and incorporates new randomly initialized parameters that enable additional input embeddings and prediction outputs. CM3leon (Yu et al., 2023) proposes an *early-fusion token-based decoder-only mixed modal* model based on the CM3 architecture that is capable of both text and image generation and editing. Unified-IO 2 (Lu et al., 2024a), Chameleon (Chameleon Team, 2024) and GPT-4o OpenAI (2024) take the early-fusion fully multimodal approach training the model from scratch to be able to expand the number of supported tasks and modalities. unlike these, which necessitate being trained from scratch using an encoder-decoder framework, GenLLaVA leverages existing pretrained models within a decoder-only architecture. This approach not only reduces computational costs but also maintains competitive performance across benchmarks.

## 3 METHOD

### 3.1 BACKGROUND: LARGE MULTIMODAL MODELS

Large language models (LLMs) excel in natural language generation, while Large Multimodal Models enhance LLMs with the ability to interpret images and respond accordingly. Built upon a pre-trained

LLM, the LMM incorporates a visual encoder (e.g., CLIP (Radford et al., 2021)) to derive visual features $f$, along with an adapter $\mathcal{W}$, usually a linear layer $\mathsf{L}$, that maps $f$ into the language domain. Following the training methodology of LLaVA (Liu et al., 2023b), this process is encapsulated in the equation:

$$
\begin{aligned}
\mathcal{X} &= \{x_1, x_2, \ldots, x_l\}, \\
f &= \mathrm{Enc}_{\mathrm{vis}}(\mathcal{V}), \\
x_t &= \mathrm{LMM}(\{x_1, \ldots, x_{t-1}\} \mid \mathcal{W}(f)),
\end{aligned}
\tag{1}
$$

where $l$ represents the number of tokens within $\mathcal{C}$. The set $\mathcal{C}$ can represent an image caption (Features Alignment) or multimodal instruction-following data (Instruction Tuning). The LMM employs the standard autoregressive method for next-token prediction, allowing it to function as a visual assistant across diverse tasks such as visual question answering and complex reasoning. We denote the next token prediction loss as $\mathcal{L}_{\mathrm{und}} = \mathrm{CE}(x_t, \mathrm{LMM}(\{x_1, \ldots, x_{t-1}\} \mid \mathcal{W}(f)))$, and it is the cross-entropy loss. Despite gaining visual perceptive abilities through this training, its responses are currently constrained to text.

### 3.2 Visual Generation in Large Multimodal Models

We append $N$ visual tokens $[\mathtt{IMG}]$ after the instruction $\mathcal{E}$, with their word embeddings being trainable. The LMM learns to generate these tokens through its language modeling (LM) head. These visual tokens represent visual-related instruction comprehension within $\mathcal{E}$ and form a bridge between the language and vision modalities. We follow the same visual generation framework of GILL (Koh et al., 2024) and MGIE (Fu et al., 2024) in extracting visual features, which we summarize here for succinctness.

We employ a generation head $\mathcal{T}$ to convert $[\mathtt{IMG}]$ into concrete visual guidance. The model $\mathcal{T}$ is a sequence-to-sequence model that translates the sequential visual tokens from the LMM into the semantically meaningful latent set $\mathcal{C} = \{c_1, c_2, \ldots, c_L\}$ for visual guidance:

$$
c_t = \mathcal{T}(\{c_1, \ldots, c_{t-1}\} \mid \{e_{[\mathtt{IMG}]} + h_{[\mathtt{IMG}]}\}),
\tag{2}
$$

where $e$ denotes the word embedding and $h$ is the hidden state (from the final layer of the LMM before the LM head) of $[\mathtt{IMG}]$. Specifically, the transformation applied to $e$ serves as a broad visual representation, while $h$ provides an instance-specific visual latent that reflects both the original image and the text conditioning the generation.

To guide image generation with the visual latent information $\mathcal{C}$, we employ a latent diffusion model (Rombach et al., 2022), incorporating a variational autoencoder (VAE) for handling denoising diffusion in the latent space. First, we encode the desired visual output via the diffusion model encoder $o = \mathrm{Enc}_{\mathrm{VAE}}(\mathcal{O})$; this output may be intended for image generation or editing tasks. The diffusion process progressively introduces noise into $o$ as $z_t$, increasing the noise level over timesteps $t$. We then train the UNet $\epsilon_\theta$ to predict the added noise (Ho et al., 2020). The diffusion process is conditioned on the visual latent information $\mathcal{C}$ through cross-attention layers, defined as $\mathrm{Attention}(Q, K, V) = \mathrm{softmax}(\frac{QK^T}{\sqrt{\mathrm{dim}}}) \cdot V$, where:

$$
Q = W_Q^{(i)} \cdot \varphi_i(z_t), K = W_K^{(i)} \cdot \mathcal{C}, V = W_V^{(i)} \cdot \mathcal{C},
\tag{3}
$$

with $\varphi$ representing the flattening operation, and $W_Q^{(i)}$, $W_K^{(i)}$, and $W_V^{(i)}$ being learnable attention matrices. The denoising score matching with latent $z_t$ is motivated similarly to Diffusion models as:

$$
\mathcal{L}_{\mathrm{gen}} = \mathbb{E}_{t \sim \mathcal{U}(0,1), \epsilon \sim \mathcal{N}(0,1)} \left[ \|\epsilon_\theta(z_t, t, |\mathcal{C}) - \epsilon\|^2 \right],
\tag{4}
$$

We apply classifier-free guidance (Ho & Salimans, 2021), where the score estimation $s_\theta$ is extrapolated to deviate from the unconditional $\varnothing$, following standard practices in diffusion models. The final loss in our model is $\mathcal{L} = \mathcal{L}_{\mathrm{und}} + 0.5\mathcal{L}_{\mathrm{gen}}$

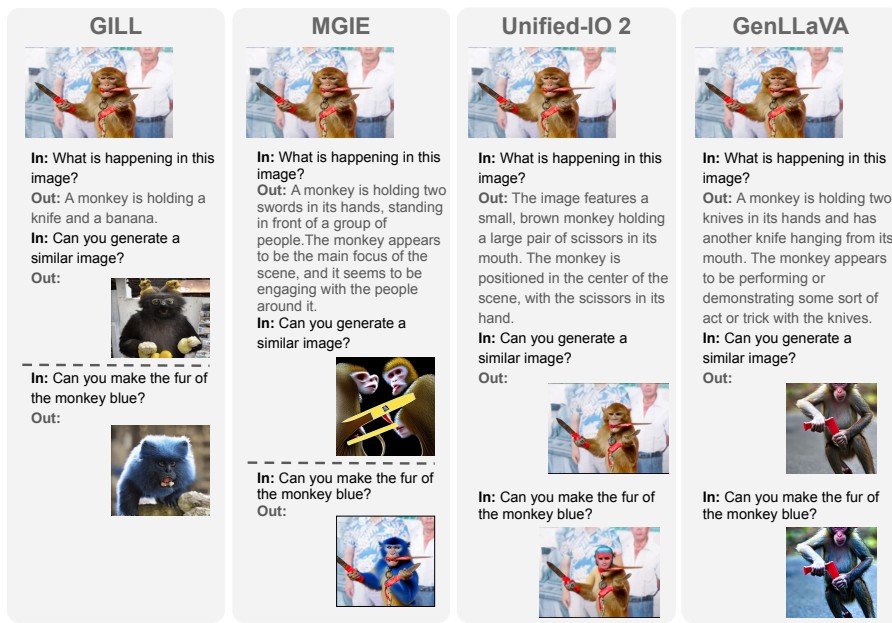

Figure 3: Qualitative conversational example of our model. The dashed line indicates that the conversation has to be restarted from the beginning due to the model losing track of it.

## 4 EXPERIMENT SETTINGS

### 4.1 GENERATIVE VISUAL INSTRUCTION DATA

Multimodal instruction tuning is a crucial process that equips the model with a wide range of skills and capabilities across different modalities while also enabling it to adapt to novel and unique instructions. We build the multimodal instruction tuning dataset by aggregating a diverse set of supervised datasets and tasks. Each task is provided with a clear prompt, either by using existing prompts or crafting new ones using GPT4-V.

**Natural Language. [1.93%]** We use the publicly available ShareGPT (ShareGPT, 2023) dataset, which was used to train the Vicuna LLM (Chiang et al., 2023). This dataset contains mostly English natural language conversations but also contains code and markdown. We filter inappropriate or low-quality entries following the same methodology of Chiang et al. (2023). As a final preprocessing step, entries that surpass 2048 tokens are truncated rather than split into multiple conversations. This results in ∼40K conversations.

**Image Editing. [9.63%]** We create a subset of the Instruction Prompt-to-Prompt dataset (IPr2Pr) (Brooks et al., 2023) for editing our editing data. We use ∼200K from the CLIP-filtered data version of IPr2Pr, where editing instructions are generated by GPT-3, and images are synthesized by the Prompt-to-Prompt model (Hertz et al., 2023).

**Image Generation. [26.88%]** For text-to-image generation, we use the same image & text pairs that were used to pre-train the LLaVA model. This dataset, named `LLaVA-Pretrain`, is inverted and presented to our model in the format (`caption`, `image`) with a dynamically pre-generated *e.g.* "*Please generate an image of* `caption`". These prefixes are created using the GPT4 language model. This dataset contains ∼558K data points originally sourced from the LAION (Schuhmann et al., 2022), SBU (Ordonez et al., 2011), and CC3M (Changpinyo et al., 2021) datasets and captioned by the BLIP-2 model (Li et al., 2023).

**Image Understanding.[61.56%]** For image understanding, we combine the dataset used to fine-tune the LLaVA model. This dataset, named `LLaVA-Finetune`, contains ∼665K samples. We also add the `LVIS-INSTRUCT4V` (Wang et al., 2023a) dataset—a new visual instruction tuning dataset constructed in the same way as the original LlaVA dataset but using GPT4-V (Achiam et al., 2023) as the captioner instead of BLIP-2 (Li et al., 2023). We remove duplicates from the resulting dataset; this results in ∼880K samples. We additionally add the following instruction datasets:

- `LRV-Instruction` (Liu et al., 2023a) ($\sim$80K) a diagram undestanding and hallucination reduction dataset.
- `laion-gpt4v-dataset` ($\sim$15K) a subset of the LAION (Schuhmann et al., 2022) dataset with high-quality captions created using GPT-4V (Achiam et al., 2023).
- `ShareGPT4V` (Chen et al., 2024a) ($\sim$100K) a conversational dataset created using publicly available conversations that users had with the GPT-4V model.
- Datasets for documents, chart and OCR understanding such as DocVQA (Mathew et al., 2021)($\sim$50K), SynDog-EN (Kim et al., 2022)($\sim$65K), ChartQA (Masry et al., 2022)($\sim$23K), DVQA (Kafle et al., 2018) ($\sim$50K) and AI2D (Kembhavi et al., 2016) ($\sim$15K).

## 4.2 Training details.

In this section, we evaluate our model on a broad range of tasks that require visual understanding and generation. **We do not perform task-specific finetuning in any experiments**. The Supplementary section details additional results on GenLLaVA 's instruction capabilities.

We adopt LLaVA-v1.5-7B (Liu et al., 2024a) architecture, then tune it on the constructed GVIT-mix-2076K. We named this model GenLLaVA and it is made of the following components:

- **Image Processing & Visual Representations.** We implement all image processing logic using the default image transforms provided by `torchvision` and the TIMM library (Wightman, 2019). We normalize pixel values using the default ImageNet values. The default backbone employed by all visual representations Enc$_{\text{vis}}$ that we evaluate in this work is a Vision Transformer (Dosovitskiy et al., 2021); we extract patch features from the *penultimate* layer, following LLaVA (Liu et al., 2023b).
- **Vision-Language Projector.** We use a simple 2-layer GELU MLP as the projector $\mathcal{W}$, which projects each patch independently into the embedding space of the language model.
- **Language Model.** We choose the Mistral-7B LLM (Jiang et al., 2023). In order to combine the projected visual patch embeddings, we perform simple sequence-wise concatenation, placing the patch embeddings before the text embeddings.
- **Visual Generation Head.** The generation head $\mathcal{T}$ is a lightweight 4-layer encoder-decoder Transformer, which takes word embeddings $e$ and hidden states $h$ from the `[IMG]` tokens, as well as $L$ learnable query tokens as the input and generates the visual latent $\mathcal{U}$, we use the $L = 77$, and the dimension of each $u_t \in \mathcal{U}$ is 768.
- **Diffusion Image Decoder.** We adopt Stable Diffusion v1.4 (SDv1.4) (Rombach et al., 2022) trained on 512×512 resolution. Similar to the visual encoder, the SD model is frozen without any modifications or training throughout the whole process.

We implement our training codebase in PyTorch. We train all models in BF16 mixed precision. For a fair comparison, the rest of the model training protocol is kept unchanged from the original LLaVA. Generative Visual Instruction tuning takes about 48 hours for both full-parameter tuning and LoRA tuning on 8 NVIDIA Tesla A100 GPUs, each with 48GB memory, with DeepSpeed ZeRO Stage 3 (Rajbhandari et al., 2020) to distribute training across GPUs.

**Single-stage training.** Unlike its predecessor LLaVA (Liu et al., 2023b), our model does not use a two-stage training pipeline and instead directly finetunes the Vision-Language projector, the Language model, and the Visual Generation Head. We found that the logical extension of the original pipeline— a three-stage training pipeline—consisting of (1) multimodal alignment, (2) instruction tuning, and (3) image generation tuning to teach the models progressively simply does not work as shown in Table 2d where we tried two different variations of the original pipeline. The model performance on visual understanding tasks decreases significantly. We instead choose a single-stage pipeline as it has been shown to work previously in the work of Karamcheti et al. (2024). This comes with other unintentional advantages: training cost is reduced by 20%, and we can use the `LLaVA-Pretrain` data as image-to-text data instead of having to collect more.

**Task Tokens.** To perform unified learning on multimodal understanding and generation, we introduce special tokens, which we name task tokens, to format the data. Specifically, we create the tokens `[T2I]` and `[I2T]` to indicate the learning (generation or understanding) task for the input sequence.

Table 1: **Main result.** Comparison of various models across advanced knowledge and general understanding. ⋆ MGIE was not originally designed for these tasks, as it is purely an editing model. For VQA, we take the generated caption as the answer, and when asking it to generate entirely new images, we provide a blank image as the prompt. We intend to show that models lose previous capabilities when we add a new one.

| Model name | Adv. Knowledge | | General Underst. | | | Editing | Generation | |
|---|---|---|---|---|---|---|---|---|
| | MathVista | MMMU | MMVet | SEED-B | MMB | EVR | CC3M | COCO |
| Emu2-34B Sun et al. (2024a) | **30.5** | 35.0 | 31.0 | **68.9** | 63.6 | - | 11.7 | 0.68 |
| Chameleon-30B Sun et al. (2024a) | 23.6 | **38.8** | 9.7 | 48.5 | 32.5 | - | 7.9 | 0.81 |
| Emu-13B Sun et al. (2024b) | 28.1 | 35.5 | 26.3 | 61.6 | 57.4 | - | 12.4 | 0.67 |
| SEED-X-17B Ge et al. (2024) | - | 35.6 | - | - | 77.8 | - | 15.0 | - |
| SEED-LLaMA-13B Ge et al. (2023) | - | - | - | 53.7 | - | - | - | 0.70 |
| Show-o-1.3B Xie et al. (2024) | - | 27.4 | - | - | - | - | 9.2 | - |
| Janus-1.3B Wu et al. (2024) | - | 30.5 | 34.3 | 63.7 | **69.4** | - | 8.5 | - |
| GILL-7B (Koh et al., 2024) | 18.6 | 26.8 | 13.0 | 29.4 | 38.2 | 30.4 | 15.3 | 0.67 |
| AnyGPT-7B (Zhan et al., 2024) | 24.4 | 24.0 | 14.8 | 28.0 | 36.0 | 40.3 | 14.3 | 0.65 |
| MGIE-7B⋆ (Fu et al., 2024) | 15.5 | 25.6 | 13.0 | 28.8 | 6.6 | **71.5** | 13.6 | 0.66 |
| Chameleon-7B (Chameleon Team, 2024) | 22.3 | 22.4 | 10.3 | 30.5 | 15.4 | - | **10.2** | **0.78** |
| DreamLLM-7B Dong et al. (2024) | - | - | 35.9 | - | 49.9 | - | 8.5 | 0.79 |
| Unified-IO 2-7B (Lu et al., 2024a) | 28.3 | 35.5 | 36.6 | 61.6 | 57.9 | 50.2 | 13.4 | 0.72 |
| GenLLaVA-7B (Ours) | **30.5** | 37.1 | **35.8** | 64.5 | 66.8 | 66.9 | 12.5 | 0.73 |

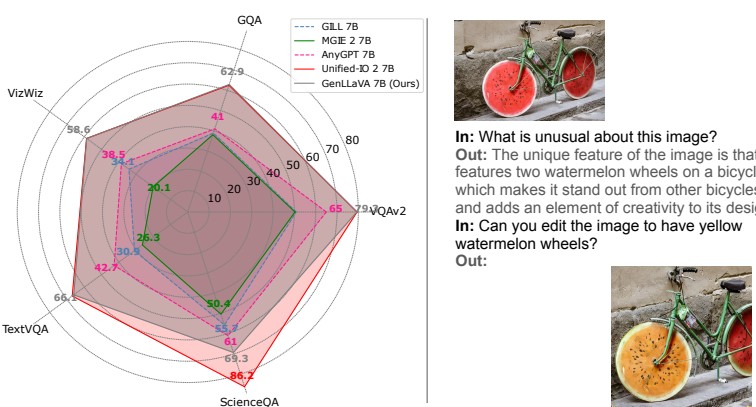

Figure 4: (Left) Results on selected Visual Question answering datasets. (Right) A qualitative example of our model.

We keep the original special token from the language model that indicates the start and end of the text. Similarly, [SOI] and [EOI] are pre-defined special tokens marking the start and end of visual tokens for generation. Without these task tokens, the model has trouble inferring the user intention and would generate an image when it is not necessary. We remark that this methodology is not new and has been used before by others (Lu et al., 2024a; Xie et al., 2024).

### 4.3 EVALUATION DETAILS.

**Visual Understanding** We evaluate vision-language performance and compare it against other generalist models, i.e., models capable of visual generation and understanding. Results on five benchmarks are shown in Table 1, designed to assess advanced knowledge and general understanding. MMBench (Liu et al., 2024c) tests answer robustness by shuffling multiple-choice options. SEED-Bench (Li et al., 2024) evaluates performance on images using multiple-choice questions. MM-Vet (Yu et al., 2024) examines visual conversational skills and response helpfulness. Mathvista (Lu et al., 2024b) probes math reasoning tests, focusing on logic and algebra. MMMU (Yue et al., 2024) spans 57 subjects from elementary to advanced levels, testing knowledge and problem-solving.

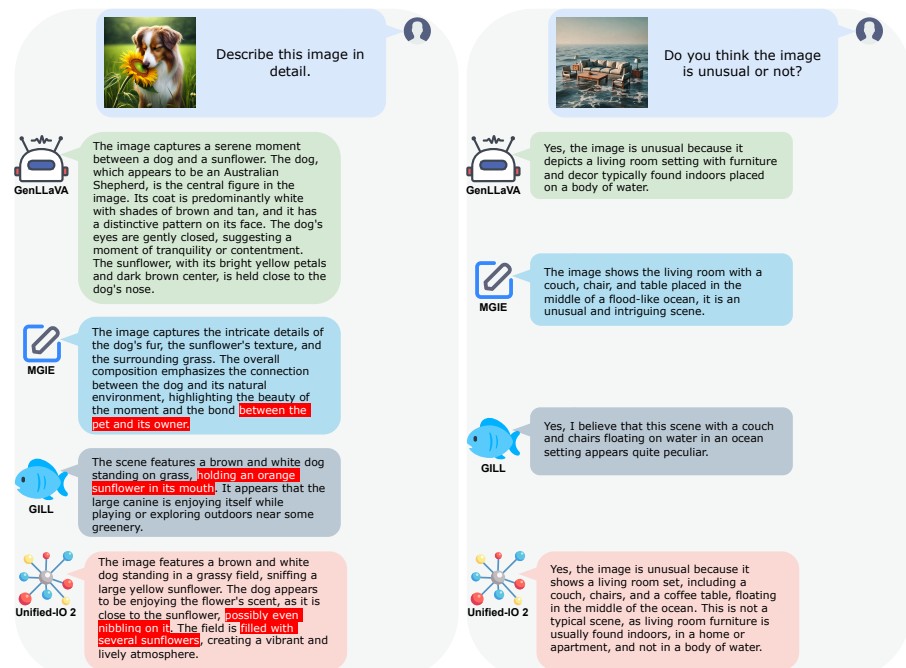

Figure 5: Comparisons of VQA capabilities among GenLLaVA, Unified-IO 2, MGIE, and GILL. One can observe that GenLLaVA is able to describe the image in detail and respond to commonly asked questions, even addressing the unusual aspects within an image. Hallucinations made by the models are highlighted in red.

**Visual Generation.** For visual generation, we evaluated Fréchet Inception Distance (FID) (Heusel et al., 2017) on the CC3M validation set (Changpinyo et al., 2021) (image realism) and CLIP Similarity on MS-COCO (Lin et al., 2014) (alignment of text prompts and generated images). For image editing, we measured DINOScore on the 5.7K EVR validation set (Tan et al., 2019), following the protocol in Fu et al. (2024).

## 4.4 MAIN RESULT

We evaluate GenLLaVA against a broad range of models across tasks involving advanced knowledge, general understanding, editing, and generative capabilities, as summarized in Table 1. The results demonstrate GenLLaVA's robust performance despite its compact size of 7B parameters, often outperforming larger models. In advanced knowledge tasks such as MathVista and MMMU, GenLLaVA achieves the highest scores, matching the performance of significantly larger models like Emu2-34B and demonstrating its superior mathematical reasoning capabilities. For general understanding, GenLLaVA excels in datasets like SEED-B and MMB, showcasing its ability to generalize effectively across diverse scenarios. It consistently surpasses competitors such as Unified-IO 2 and Emu-13B on these benchmarks.

Although specialized editing models like MGIE have a marginal advantage in image editing (EVR), GenLLaVA maintains competitive performance, demonstrating its adaptability as a generalist model. In visual generation, GenLLaVA achieves high alignment between generated images and prompts, with results closely matching Unified-IO 2 on CC3M and COCO benchmarks. Overall, GenLLaVA balances multimodal capabilities without compromising performance in individual domains. Its consistent strength across advanced reasoning, general understanding, and generative tasks emphasizes its versatility and potential as a robust vision-language model.

## 4.5 RESULTS ON SELECTED VISUAL QUESTION ANSWERING DATASETS.

We evaluate various models' performance across diverse visual question-answering datasets, including VQAv2, GQA, VizWiz, TextVQA, and ScienceQA. Our results show that Unified-IO 2 and GenLLaVA consistently perform well across most datasets. Specifically, Unified-IO 2 achieves the

highest scores on the ScienceQA (86.2%) and TextVQA (67%), while GenLLaVA demonstrates strong performance on VQAv2 (79.3%) and a competitive score on GQA (62.9%). In contrast, GILL and MGIE exhibit generally lower performance across all datasets, with MGIE notably struggling on VizWiz (20.1%) and TextVQA (26.3%). AnyGPT shows moderate effectiveness, with its best performance on ScienceQA (61%). We used VLMEvalKit from Duan et al. (2024) to get the results for these datasets, which perform a generation-based evaluation using the LLM-as-a-judge protocol.[1] The results can be seen in Fig. 4 and Fig. 5.

## 4.6 ABLATIONS

We investigate the effect of scaling the data used to train GenLlaVA, the effect of using different image backbones, and the number of visual tokens used for image generation.

**Instruction data.** We start with the original instruction tuning dataset from LlaVA-1.5, basically reproducing the original results using a one-stage training recipe. We add the `LLaVA-Pretrain` and generation head to our model, and we notice that adding generation capabilities significantly affects the visual understanding capabilities, with all metrics degrading between 2.3% (MM-Vet) and 7.9% (MathVista). To compensate for this loss in performance, we modify the ratio of image generation to image understanding data in our dataset from ∼50%-50% to ∼70%-30%, by adding more image instruction data from `LVIS-INSTRUCT4V`, `LRV-Instruction` and other chart understanding datasets. This results in a model with significant generation capabilities that maintain its image-understanding capabilities. We finally add image-generation capabilities using our selected subset of the IPr2Pr dataset. This reduces the image understanding capabilities, but we consider this a small enough change that balances the three tasks while maintaining commendable performance. Finally, task tokens are added to the resulting instruction set to condition the model on the desired task. The results can be seen in Table 2a.

**Choice of Visual encoder.** The quality of the vision encoder can also have big effects on the final LMM performance. We start by using a CLIP/B, which has the lowest performance, then we compare against a stronger visual encoder and create versions of GenLLaVA, which is trained with CLIP (Radford et al., 2021) and SigLIP (Zhai et al., 2023), respectively. We can see in Table 2b that the SigLIP encoder generally achieves better performance than the CLIP encoder. This shows that SigLIP is a better vision encoder for LMM development.

**Number of visual generation tokens.** We experiment with varying the number of visual generation tokens, $N$, to determine the optimal number required for balancing image generation and understanding tasks. We noticed that we need more visual generation tokens than GILL ($N = 4$) and MGIE ($N = 8$) to achieve the best performance. We find that $N = 16$ is the best choice for our model. We hypothesize that this is because our model has to balance image generation and editing in the same head and thus needs more visual tokens to capture the complexity of the tasks. The results can be seen in Table 2c.

## 4.7 COMPARISON WITH THE STATE OF THE ART.

When compared with state-of-the-art models, GenLLaVA maintains similar performance to models of the LLaVA family when evaluated using the average of the scores on the MathVista, MMMU, MMVet, SEED-B, and MMB datasets. However, it lags behind larger and more specialized models. However, some of these models lack the generative capabilities present in GenLLaVA.

When compared with models of similar size and setup (∼ 7b parameters), our model surpasses the original LlaVAv1(Liu et al., 2023b) model by 9% points (37.5% vs. 46.9%), and LlaVA-1.5 (Liu et al., 2024a) by ∼1% points (45.3% vs. 46.9%). It is surpassed by the LlaVA-Next (Liu et al., 2024b) family of models by ∼4% points; by Idefics2 (Laurençon et al., 2024) by ∼8% points (55.7% vs. 46.9%). Compared with the absolute state-of-the-art open-source models, GenLLaVA lags behind Yi-VL (Young et al., 2024) by ∼2% points; and surpasses Emu2 (Sun et al., 2024a) by ∼1% points. It lags behind LlaVA-NeXT (Liu et al., 2024b) (34b) by ∼12% points, and InternVL 1.5 (Chen et al., 2024b) by ∼15% points. Compared with the absolute state-of-the-art closed models, GenLLaVA lags behind GPT-4o by ∼26% points, GPT-4V (Achiam et al., 2023) by ∼20% points; and the Gemini family (Gemini Team et al., 2023) of models by 13% points.

---

[1]We used GPT-4 (0409) as the judge.

Table 2: **Ablation experiments** on different datasets. We present evaluation results across various ablation types.

(a) **Data Ablation**. We study the effect of progressively adding data to the model, starting with visual understanding-only data and then incorporating a mix of both understanding and generation tasks.

| Model Variation | Advanced Knowledge | | General Understanding | | | Editing | Generation | |
|---|---|---|---|---|---|---|---|---|
| | MathVista | MMMU | MM-Vet | SEED-B | MMB | EVR | CC3M | COCO |
| LLaVA-Finetune | 24.7 | 28.7 | 30.1 | 54.7 | 65.6 | - | - | - |
| Generation | 16.8 | 27.5 | 27.8 | 52.1 | 59.8 | 30.2 | 13.9 | 0.73 |
| Extra Knowledge | 28.2 | 31.8 | 32.4 | 59.7 | 64.1 | 28.5 | 14.0 | 0.72 |
| IPr2Pr-200K | 24.9 | 29.7 | 33.1 | 63.5 | 65.0 | 64.7 | 14.3 | 0.71 |
| Task Tokens | **30.5** | **37.1** | **35.8** | **64.5** | **66.8** | **66.9** | **12.5** | **0.73** |

(b) **Vision Encoder Ablation**. We study the performance of using different vision encoders across several visual understanding and generation benchmarks. We do not condition on the task tokens for this experiment.

| Model Variation | Advanced Knowledge | | General Understanding | | | Editing | Generation | |
|---|---|---|---|---|---|---|---|---|
| | MathVista | MMMU | MM-Vet | SEED-B | MMB | EVR | CC3M | COCO |
| CLIP/B-224px | 23.6 | 28.6 | 29.1 | 53.4 | 60.3 | 62.1 | 15.0 | 0.68 |
| CLIP/L-336px | 24.6 | 29.2 | 32.4 | 59.7 | 64.1 | 64.5 | 14.4 | 0.70 |
| SigLIP/L-384px | **24.9** | **29.7** | **33.1** | **63.5** | **65.0** | **64.7** | **14.3** | **0.71** |

(c) **Number of Generation Tokens**. We study the performance across several visual understanding and generation benchmarks when varying the number of visual generation tokens (denoted as $N$). We do not condition on the task tokens for this experiment.

| Model Variation | Advanced Knowledge | | General Understanding | | | Editing | Generation | |
|---|---|---|---|---|---|---|---|---|
| | MathVista | MMMU | MM-Vet | SEED-B | MMB | EVR | CC3M | COCO |
| $N = 4$ | **30.7** | **35.0** | 30.5 | 58.2 | 64.7 | 40.3 | 17.0 | 0.62 |
| $N = 8$ | 28.7 | 32.8 | 32.4 | 61.2 | **65.3** | 53.3 | 15.6 | 0.67 |
| $N = 16$ | 24.9 | 29.7 | **33.1** | **63.5** | 65.0 | **64.7** | **14.3** | **0.71** |

(d) **Recipe Ablation**. We compare different training recipes to evaluate their impact on model performance across advanced knowledge, general understanding, and generation tasks.

| Recipe Type | Advanced Knowledge | | General Understanding | | | Editing | Generation | |
|---|---|---|---|---|---|---|---|---|
| | MathVista | MMMU | MMVet | SEED-B | MMB | EVR | CC3M | COCO |
| Gen. first → Und. last | 15.5 | 29.6 | 13.0 | 44.5 | 32.9 | - | 15.9 | 0.64 |
| Und. first → Gen. last | 14.6 | 23.6 | 8.3 | 25.1 | 20.8 | - | 14.3 | 0.66 |
| Und. Only | 34.0 | 41.0 | 37.4 | 68.9 | 68.9 | - | - | - |
| GenLLaVA | **30.5** | **37.1** | **35.8** | **64.5** | **66.8** | **66.9** | **12.5** | **0.73** |

# 5 CONCLUSION

In this paper, we have introduced the Generative Large Language and Visual Assistant (GenLLaVA), a comprehensive framework for enabling Large Multimodal Models (LMMs) to excel simultaneously in image understanding, generation, and editing, while maintaining competitive performance. By balancing multimodal capabilities within a single model through the curation of a diverse multimodal instruction dataset and the development of an innovative single-phase training methodology, GenLLaVA sets a new benchmark in the development of multimodal systems. Our results show that unifying generation and understanding under a single framework is possible without compromising their strengths. Our work sets a new standard for building visual assistants with extra capabilities, and we hope our open-source contributions, including datasets, codebase, and model checkpoints, will serve as valuable resources for the research community, driving further advancements in the field of multimodal AI. GenLLaVA's capabilities can be extended to video understanding, audio-visual tasks, and more advanced real-time multimodal interactions.

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

**System Prompt:**

A chat between a curious user and an artificial intelligence assistant. The assistant gives helpful, detailed, and politely answers to the user's questions.

- - - - - - - - - - - - - - - - - - - - - - - - - - - - - - - - - - - - - - - - - - - -

(a) VQA (Short)

USER: Based on the image, please answer the question. <IMAGE> <QUESTION> Please provide an accurate answer within one word.
ASSISTANT: The answer is: <ANSWER>

- - - - - - - - - - - - - - - - - - - - - - - - - - - - - - - - - - - - - - - - - - - -

(b) VQA (Long)

USER: This is an exam, please answer according to the image and question. <IMAGE> <QUESTION>
ASSISTANT: The answer is: <ANSWER>

- - - - - - - - - - - - - - - - - - - - - - - - - - - - - - - - - - - - - - - - - - - -

(c) ADVANCED KNOWLEDGE

USER: This is a hard exam, please answer according to the image and the question. <IMAGE> <QUESTION> Please think step by step.
ASSISTANT: The answer is: <ANSWER>

- - - - - - - - - - - - - - - - - - - - - - - - - - - - - - - - - - - - - - - - - - - -

(d) GENERATION

USER: Generate an image with the following description. <DESCRIPTION>
ASSISTANT: <GENERATED IMAGE>

- - - - - - - - - - - - - - - - - - - - - - - - - - - - - - - - - - - - - - - - - - - -

(e) EDITING

USER: Based on the image, please follow the instruction. <IMAGE> Edit the image according to the description. <DESCRIPTION>
ASSISTANT: <GENERATED IMAGE>

Figure A1: **Prompt templates**. (a) Short VQA includes VQAv2, VizWiz, GQA, and TextVQA and ScienceQA. (b) Long VQA includes MMB, SEED-B, and MM-Vet. (c) Advanced knowledge includes MathVista and MMMU. (d) Generation includes CC3M and MS-COCO. (e) Editing includes EVR. <IMAGE> is the image representation, <QUESTION> denotes each specific question, <ANSWER> is the generated answer, <DESCRIPTION> is an image description for generation or editing, and <GENERATED IMAGE> is the output of the generation head..

## A APPENDIX

### A.1 IMPLEMENTATION DETAILS

We adopt the hyperparameters in Table A1 for all our experiments.

Table A1: **Training Hyperparameters**

| Hyperparameter | Value |
|---|---|
| Batch Size | 128 |
| Max Gradient Norm | 1.0 |
| Weight Decay | 0.1 |
| Learning Rate | 2e-5 |
| Optimizer | AdamW |
| Scheduler | Warmup & Cosine Decay |
| Warmup Ratio | 0.03 |

### A.2 EVALUATION DETAILS

During the evaluation with VLMEvalKit (Duan et al., 2024), we used the prompt templates listed in Fig A1

### A.3 ADDITIONAL QUALITATIVE EXAMPLES

In Tables A2 and A3, we present a qualitative comparative analysis of VQA results between our model, GenLLaVA, and other state-of-the-art models: GPT-4 (Achiam et al., 2023), LLaVA (Liu et al., 2024a), GPT-4o (OpenAI, 2024), and InternVL-1.5 (Chen et al., 2024b). Our model, while smaller than state-of-the-art models, is still able to give detailed and precise responses to given questions, avoiding the introduction of hallucinations, unlike LLaVA-1.5.

**Visual input example, Extreme Ironing:**

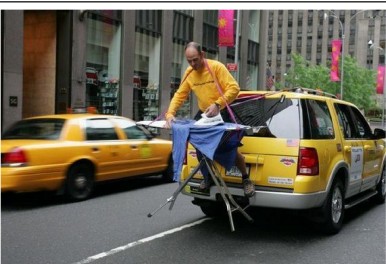

Source: `https://www.barnorama.com/wp-content/uploads/2016/12/03-Confusing-Pictures.jpg`

| | |
|---|---|
| User | What is unusual about this image? |
| GenLLaVA (Ours) | The unusual aspect of this image is that a man is ironing clothes on the back of a yellow SUV in the middle of a city street. This is not a common sight, as ironing is typically done indoors, in a private setting. The presence of a taxi cab in the background further emphasizes the unexpected nature of the scene, as it juxtaposes the mundane task of ironing with the hustle and bustle of urban life. |
| User | What is unusual about this image? |
| LLaVA (Liu et al., 2024a) | The unusual aspect of this image is that a man is ironing clothes while standing on the back of a moving car. This is not a typical scene, as ironing clothes is usually done indoors, in a stationary position, and not while the vehicle is in motion. The man's actions are both unconventional and potentially dangerous, as ironing clothes on a moving car can be hazardous and may cause accidents or injuries. |
| User | What is unusual about this image? |
| GPT-4V (Achiam et al., 2023) | The unusual aspect of this image is the man ironing on a portable ironing board mounted to the back of a yellow taxi in the middle of a bustling city street. This activity is not typically associated with such a setting, making it an extraordinary and humorous sight. This represents an extreme form of "extreme ironing," a performance sport where people iron clothes in unexpected and often difficult locations. |
| User | What is unusual about this image? |
| GPT-4o (OpenAI, 2024) | The image shows a man ironing clothes on an ironing board that is attached to the back of a moving vehicle, which appears to be a yellow taxi. This is highly unusual because ironing is typically done indoors, on stable surfaces, and definitely not while attached to a moving car. This extreme scenario might be an example of "extreme ironing," a humorous and unconventional sport where people iron clothes in extreme or unusual situations. The image itself blends an ordinary chore with a risky and absurd setting, making it quite striking and humorous. |
| User | What is unusual about this image? |
| InternVL-1.5 (Chen et al., 2024b) | This image depicts an unusual scene in which a person is ironing clothes on an ironing board attached to the back of a taxi while it is parked on a city street. Several elements make this situation peculiar: (1) Location: Ironing is typically done indoors at home or in a laundry room where there are proper facilities for such tasks. (2) Practicality: It's impractical to set up an ironing board and use an iron outdoors, especially in a busy urban environment where distractions and interruptions could occur frequently. The juxtaposition of a mundane domestic activity with the bustling cityscape creates a humorous and unexpected scenario. |

Table A2: **Visual question answering example comparison** of GenLLaVA to LLaVA-1.5-7B(Liu et al., 2024a), GPT-4V (Achiam et al., 2023), GPT-4o (OpenAI, 2024), and InternVL-1.5 (Chen et al., 2024b). This table follows the format from Bubeck et al. (2023).

**Visual input example, Chicken Nugget Map:**

Sometimes I just look at pictures of the earth from space and I marvel at how beautiful it all is.

| | |
|---|---|
| User
GenLLaVA (Ours) | Can you explain this meme in detail?
The meme is a play on the phrase "beauty is in the eye of the beholder," which means that what one person finds beautiful may not be the same for another. In this case, the person creating the meme finds beauty in the shape of the Earth formed by pieces of fried chicken arranged on a baking tray. It's a humorous take on appreciating simple things and finding artistic or aesthetic value in everyday objects or food items. |
| User
LLaVA (Liu et al., 2024a) | Can you explain this meme in detail?
The meme in the image is a creative and humorous take on food, with a focus on chicken nuggets, as the center of the universe. The meme begins with the phrase "Sometimes I just look at pictures of the Earth from space and I marvel at how beautiful it all is..." The main image features a tray of cooked chicken nuggets, with the chicken pieces positioned to resemble the continents and islands on a world map. The meme playfully suggests that the chicken nuggets represent the Earth, and the various locations depicted in the photo are actually chicken nugget versions of different places. The punchline of the meme appears in smaller text at the bottom, reading "I mean, it's not the real Earth, but how beautiful it is all is." This lighthearted and entertaining meme combines food and humor, using the image of chicken nuggets to create a funny and imaginative representation of the world. |
| User
GPT-4V (Achiam et al., 2023) | Can you explain this meme?
This meme cleverly uses chicken nuggets shaped like continents to make a humorous comparison. The caption suggests that the person admires pictures of Earth from space for their beauty, but the image shows chicken nuggets on a baking tray arranged to look like a world map. This juxtaposition creates a playful contrast between the expected awe-inspiring images of our planet and the mundane, yet amusing arrangement of food. The humor lies in the unexpected substitution and the notion that even chicken nuggets can evoke a sense of wonder if seen from the right perspective. |
| User
GPT-4o (OpenAI, 2024) | Can you explain this meme in detail?
This meme humorously compares the shapes of fried chicken nuggets to a map of the world, as though the person is marveling at the beauty of the Earth from space. The chicken nuggets are arranged in a way that resembles the continents, and the caption plays off the idea of someone gazing at these shapes with awe, as if they're looking at Earth, rather than just pieces of food.
The humor comes from the exaggerated comparison of something mundane (fried chicken nuggets) to something grand and majestic (the Earth viewed from space). It's a playful and lighthearted take on how our brains often find patterns in random objects that resemble familiar things. |
| User
InternVL-1.5 (Chen et al., 2024b) | Can you explain this meme in detail?
This meme humorously contrasts the profound and contemplative act of admiring the beauty of Earth from space with a more mundane, everyday activity. The top part of the image includes text that says: "Sometimes I just look at pictures of the earth from space and I marvel at how beautiful it all is." Following this statement, instead of showing an image of Earth from space, there is a picture of several pieces of fried chicken arranged on a baking sheet.
The joke lies in the unexpected and humorous twist, where the "beautiful sight" referenced in the text turns out to be something as ordinary and relatable as fried chicken. This creates a light-hearted contrast between grandeur (Earth viewed from space) and simplicity (fried chicken), playing on the idea that both can evoke similar feelings of wonder or satisfaction. |

Table A3: **Visual question answering example comparison** of GenLLaVA to LLaVA-1.5-7B(Liu et al., 2024a), GPT-4V (Achiam et al., 2023), GPT-4o (OpenAI, 2024), and InternVL-1.5 (Chen et al., 2024b). This table follows the format from Bubeck et al. (2023).

