# OpenReview forum: "Generative Visual Instruction Tuning"
_ICLR.cc/2025/Conference — Submitted to ICLR 2025_

### Official Review · Reviewer_B4pt · 2024-10-18

**Soundness:** 2
**Presentation:** 2
**Contribution:** 2
**Rating:** 3
**Confidence:** 5

**Summary:**

This paper introduces a multimodal instruction tuning set that combines image understanding, image generation, and image editing data. It further introduces a model GenLLaVA, which unifies visual understanding and generation using open-source models.

**Strengths:**

1. The authors build a multimodal instruction tuning dataset by aggregating a diverse set of supervised datasets and tasks including text understanding, image comprehension, generation and editing.
2. The paper is generally easy to follow.

**Weaknesses:**

1. Unified multimodal understanding and generation has become a highly popular direction recently, with many significant models introduced this year that can achieve image understanding, generation, and editing. However, this method simply adopts the same framework as GILL, which translates the hidden representations of an LLM into text embeddings of a pre-existing stable diffusion model. The mention of "using a Q-former as the generation head" in the paper cannot be considered a distinguishing feature from the GILL approach. From a methodological standpoint, this work appears to lack any innovation.
2.  This paper only compares a few outdated and underperforming models as baselines. The recent multimodal LLMs with unified comprehension and generation should be compared including Emu [1], Emu-2.0 [2], SEED-LLaMA [3], SEED-X [4], Mini-Gemini [5], etc.
3. The claim that "To our knowledge, this is the first time such capability has been achieved" is unfounded. As I mentioned earlier, several works have already accomplished unified image understanding, generation, and editing.

[1] Generative pretraining in multimodality
[2] Generative multimodal models are in-context learners
[3] Making llama see and draw with seed tokenizer
[4] SEED-X: Multimodal Models with Unified Multi-granularity Comprehension and Generation
[5] Mini-Gemini: Mining the Potential of Multi-modality Vision Language Models

**Questions:**

Please see weaknesses.

---

> ### Author Response · Authors · 2024-11-21
> **Response to Reviewer B4pt**
>
> We thank the reviewer for the feedback and comments. Below are our responses.
>
> **Limited Innovation**
>
> We emphasize the following to be our key contributions:
>
> - **Model Representation** : Unlike GILL, which maps LLM embeddings to text embeddings of the Stable Diffusion model, our method directly injects LLM representations into the visual latent space of the diffusion process. This distinction allows GenLLaVA to enable image editing capabilities directly influenced by visual context rather than textual abstraction.
>
> - **Model Architecture** : Our approach is distinct from Unified-IO 2 (Encoder-Decoder) as it adopts a decoder-only architecture, reuses existing models, and does not require training from scratch, thereby significantly reducing computational costs.
>
> - **Dataset Contribution** : We compiled a novel fine-tuning dataset that integrates multimodal instruction tuning data for image understanding, generation, and editing. Unlike existing works, this dataset enables researchers to bypass the need for collecting and combining individual datasets manually.
>
> - **Q-Former Utilization** : While superficially similar to prior work, the Q-former enables better integration with latent diffusion and improves generative performance, as reflected in our ablations in Section 4.6.
>
> ---
>
> **Baseline Comparison**
>
> We appreciate the reviewer's suggestions regarding the choice of baselines and have expanded the evaluation to include more models, such as Emu, SEED-X, Show-O, and DreamLLM in the following table:
>
>
> | Model Name| MathVista | MMMU | MMVet | SEED-B | MMB  | EVR  | CC3M  | COCO  |
> |---------------|-----------|------|-------|-----------|-------|-------|-------|-------|
> |**30B Models**|           |      |       |        |       |      |       |       |
> |Emu2| 30.5      | 35   | 31    | 68.9   | 63.6  | -    | 11.7  | 0.68  |
> |Chameleon| 23.6      | 38.8 | 9.7   | 48.5   | 32.5  | -    | 7.9   | 0.81  |
> |**10B Models**|          |      |       |        |       |      |       |       |
> |Emu1| 28.1      | 35.5 | 26.3  | 61.6   | 57.4  | -    | 12.4  | 0.67  |
> |SEED-X| -         | 35.6 | -     | -      | 77.8  | -    | 15    | -     |
> |SEED-LLaMA| -         | -    | -     | 53.7   | -     | -    | -     | 0.7   |
> |**3B Models**|          |      |       |        |       |      |       |       |
> |Show-o| -         | 27.4 | -     | -      | -     | -    | 9.2   | -     |
> |**7B Models**|         |      |       |        |       |      |       |       |
> |GILL| 18.6      | 26.8 | 13.0  | 29.4   | 38.2  | 30.4 | 15.3  | 0.67  |
> |AnyGPT| 24.4      | 24.0 | 14.8  | 28.0   | 36.0  | 40.3 | 14.3  | 0.65  |
> |MGIE| 15.5      | 25.6 | 13.0  | 28.8   | 6.6   | 71.5 | 13.6  | 0.66  |
> |Chameleon| 22.3      | 22.4 | 10.3  | 30.5   | 15.4  | -    | 10.2  | 0.78  |
> |DreamLLM| -         | -    | 35.9  | -      | 49.9  | -    | 8.5   | 0.79  |
> |Unified-IO-2| 28.3      | 35.5 | 36.6  | 61.6   | 57.9  | 50.2 | 13.4  | 0.72  |
> |GenLLaVA| 30.5      | 37.1 | 35.8  | 64.5   | 66.8  | 66.9 | 12.5  | 0.73  |
>
> **Table B**: Performance comparison of GenLLaVA against state-of-the-art models on multimodal understanding  and generation tasks. We will fill in the missing values in the table for the revised manuscript.
>
> Some models mentioned by the reviewer, such as Mini-Gemini, are not true generative models, as they rely on creating detailed prompts for a diffuser (akin to tool use).
>
> ---
>
> **Unfounded Claims**
>
> We will qualify our claims to indicate how we make progress on this problem as we are being made aware of other efforts and we contextualize our work further (see Table 1 and Section 4.7 of the manuscript for more details).
>
> ---

---

> ### Comment · Reviewer_B4pt · 2024-11-27
>
> Thanks for the authors' response. However, I still concern about the novelty of this paper considering a large number of advanced models introduced this year that can achieve unified image understanding, generation, and editing. I decided to keep my rating.

---

### Official Review · Reviewer_borP · 2024-10-31

**Soundness:** 3
**Presentation:** 3
**Contribution:** 2
**Rating:** 6
**Confidence:** 3

**Summary:**

This paper presents a new MLLM that has multi modal capabilities on both input and output. It also presents a new instruction following dataset, which is a non-trival combination and selection from a large amount of existing datasets for various specific tasks.
The paper adopts a less commonly used single-stage instruction tuning pipeline, which according to the text is more effective than extending previous multi-stage pipeline.
Result evaluation shows that the model surpasses many major MLLMs on visual understanding tasks, and is roughly on par with previous models emphasizing visual generation tasks.

The author claims that model checkpoint and the new dataset will be released.

**Strengths:**

While the task and the approach are not completely new, the paper proposed several improvements on current approach to build MLLMs, and demonstrated their effectiveness using relatively comprehensive evaluations, resulting in a new model that is strong on both visual understanding and visual generation.

The paper is largely clear about the goal, approach and results, though it will be better if more details on the reason of several design choices can be made clear.

**Weaknesses:**

The biggest weakness is the incremental nature of the work.  For example,
the paper claims it unifies visual understanding and generation, but there are similar models such as SEED-X, which is also capable of both, as well as the compared work such as Unified-IO 2.

the paper claims they contribute a dataset for instruction tuning, but the dataset is composed of multiple existing datasets. The novelty to me would be insights on why these sets are selected, and why for some datasets such as IPr2Pr only a subset is selected. The paper can be much more impactful if there are more discussions on these decisions.

**Questions:**

Similar as putting MGIE as a baseline model, which is a domain expert on image editing, it would be great if other domain expert models can be put as references as well. The proposed model doesn't have to beat these models, but it will give readers a more comprehensive understanding of how good a combined model can be compared with domain experts.

The paper acknowledges that task specific tokens such as [T2I] and [I2T] are needed to achieve this balanced performance. In real user queries these tokens are not likely to be provided, I wonder if there are other more natural approaches to improve user intent understanding.

---

> ### Author Response · Authors · 2024-11-22
> **Response to Reviewer borP**
>
> We thank the reviewer for the feedback. Below are our detailed responses.
>
> **Incremental Nature of the Work**
> We emphasize the following key distinctions from prior work, specifically SEED-X and Unified-IO 2:
>
> - **Model Training** : Unlike Unified-IO 2, which is trained from scratch with an encoder-decoder architecture, our model adopts a decoder-only architecture and reuses existing models, reducing computational costs while maintaining good performance.
>
> - **Dataset Contribution** : We compiled a novel fine-tuning dataset for multimodal tasks (understanding, generation, and editing). Unlike SEED-X, which only releases an image editing dataset, our dataset integrates these modalities seamlessly.
>
> - **Omnimodality** : SEED-X relies on task-specific checkpoints (SEED-X-I, SEED-X-Gen, SEED-X-Edit), forcing users to switch models for different tasks. While SEED-X includes a general-purpose checkpoint, it significantly underperforms across tasks. In contrast, GenLLaVA balances all modalities, achieving good performance without task-specific checkpoints.
>
> - **Generation Limitations** : SEED-X relies heavily on CLIP features for generation, which inherently are not fine-grained enough for high-quality Generation. GenLLaVA overcomes this by integrating features generated by the LLM, enabling finer control and better Generation capabilities.
>
> ---
>
> **Domain-Expert Comparisons**: Including more domain-expert baselines would provide a comprehensive perspective. We will include additional results for domain-specific models (e.g., MGIE for image editing) in Table C and the supplementary material. These results contextualize the performance of GenLLaVA as a unified model able to balance these tasks whitin the same model.
>
> | Model          | MathVista | MMMU | MMVet | SEED-B | MMB  | EVR  | CC3M | COCO |
> |----------------------|-----------|------|-------|--------|-------|------|-------|------|
> | Img. Gen Expert      | -         | -    | -     | -      | -     | -    | 11.9  | 0.75 |
> | Img. Editing Expert  | 15.5      | 25.6 | 13.0  | 28.8   | 6.6   | 71.5 | 13.6  | 0.66 |
> | Img. Und. Expert     | 34.0      | 41.0 | 37.4  | 68.9   | 68.9  | -    | -     | -    |
> | GenLLaVA             | 30.5      | 37.1 | 35.8  | 64.5   | 66.8  | 66.9 | 12.5  | 0.73 |
>
> **Table C**: Performance comparison of domain-specific experts and GenLLaVA across various tasks. Img. Gen Expert is SD1.4, Img. Editing Expert is MGIE, and Img. Und. Expert refers to a LLaVA model trained on the image understanding data we collected.
>
>
> ---
>
> **User Intent Understanding**: The use of task tokens ([T2I], [I2T]) is a compromise for balancing performance across tasks. Inferring user intent remains an open challenge, as evidenced by the limitations of purely omnimodal systems like Chameleon, which sometimes fail to distinguish between tasks effectively. Other strategies, such as modality-specific checkpoints (e.g., Emu models, SEE-X), bypass intent inference entirely but compromise generality.
>
> Our approach aligns with other state-of-the-art models, such as Unified-IO 2, Show-O, and DreamLLM, which also utilize task-specific tokens. This strategy explicitly conditions the model during training and inference, reducing task ambiguity, providing the user with a way to seamlessly change tasks without having to re-load a new chekpoint.
>
> ---

---

### Official Review · Reviewer_uJ5q · 2024-11-02

**Soundness:** 3
**Presentation:** 4
**Contribution:** 3
**Rating:** 5
**Confidence:** 5

**Summary:**

This paper proposes a composite model, GenLLaVA, for image understanding, generation, and editing. The authors create a generative multimodal instruction-following dataset using GPT-4V and existing datasets for image generation and editing. They also explore various training strategies and model designs, including single-stage training, the selection of a vision encoder, and the use of task tokens. Quantitative results across multiple benchmarks demonstrate GenLLaVA's effectiveness across all three tasks.

**Strengths:**

1.	Developing generative multimodal instruction-following data could be highly valuable for future research and applications.
2.	The model shows strong performance across image understanding, image editing, and image generation tasks.
3.	The paper is well-presented, making it easy to follow and understand.

**Weaknesses:**

1.	Although the authors promise to open-source all materials, including data, code, and pre-trained weights, none of these resources are provided for review. Since the dataset is a key contribution of this work, it would strengthen the paper to include these materials in the revised manuscript (with anonymity). I would consider recommending acceptance only if these resources are included in the final version.
2.	The novelty of this paper appears limited, as the model seems to be a straightforward combination of existing techniques, such as the framework, task tokens, vision encoder, and single-stage training.

**Questions:**

1.	What makes single-stage training preferable over multi-stage training in GenLLaVA.
2.	The performance of GenLLaVA significantly trails behind recent state-of-the-art models. For instance, the new version of LLaVA, LLaVA-OV [A], achieves an 80.8 on MMB, outperforming GenLLaVA by 14 points. I'm curious how the proposed dataset would perform if a stronger model and large-scale understanding data were used. Would a new mix of data be necessary, or would the current amount of generative data suffice?

[A] Li, B., Zhang, Y., Guo, D., Zhang, R., Li, F., Zhang, H., ... & Li, C. (2024). Llava-onevision: Easy visual task transfer. arXiv preprint arXiv:2408.03326.

---

> ### Author Response · Authors · 2024-11-23
> **Response to Reviewer uJ5q**
>
> We thank the reviewer for their comments. Below are our responses:
>
> **Limited Originality in Methodology**
>
> - **Model Design and Contributions** :
>   - Our model is a decoder-only architecture that reuses existing pre-trained components (Mistral LLM, SigLIP vision encoder, and Stable Diffusion), which reduces computational requirements compared to training from scratch (e.g., Unified-IO 2).
>
>   - Unlike prior works such as GILL, which guide the diffusion model solely via text prompts, GenLLaVA directly feeds features from the LLM into the U-Net visual encoder of the diffusion model. This enables direct visual guidance during diffusion, facilitating tasks like image editing. In contrast, text-only guidance (e.g., MiniGemini) necessitates intermediate text conversion, limiting editing capabilities.
>
>   - Models like SEED-X and the Emu family require task-specific checkpoints (SEED-X-I, SEED-X-gen, SEED-X-edit) for understanding, generation, and editing, reducing generalization. GenLLaVA maintains a unified model with competitive performance across tasks, providing the user with a way to seamlessly change tasks without having to re-load new checkpoints.
>
> - **Dataset Contributions** :
>   - We curated a multimodal instruction-following dataset that extends LLaVA-Finetune with new generative and editing tasks, offering a comprehensive, reusable resource for the community.
>
>
> ---
>
> **Preference for Single-Stage Training**
> Our decision for a single-stage training pipeline stemmed from the following observations:
>
> - **Empirical Performance** :
>   - Multi-stage pipelines, including recipes that separate multimodal alignment, generative capability addition, and instruction tuning, resulted in catastrophic forgetting. For example, when multimodal alignment was conducted first, subsequent stages significantly degraded the model’s understanding performance (see Table 1 in the manuscript for results).
>
>   - Single-stage training, as shown in prior works like PrismaticVLMs, simplifies the process while maintaining competitive results for both understanding and generative tasks. We remark that, we only cite PrismaticVLMs, as the first to do so but only for image understanding tasks, while we extend this to image generation and editing tasks.
>
> - **Practical Advantages** :
>   - Single-stage training reduces computational costs by approximately 20% and enables us to leverage existing datasets like LLaVA-Pretrain without the need for additional alignment data collection.
>
> We have added comparisons with different training strategies to Table D for further clarity.
>
> | Recipe Type              | MathVista | MMMU | MMVet | SEED-B | MMB  | EVR  | CC3M | COCO |
> |--------------------------|-----------|------|-------|--------|-------|------|-------|------|
> | Gen. first -> Und. last  | 15.5      | 29.6 | 13.0  | 44.5   | 32.9  | -    | 15.9  | 0.64 |
> | Und. first -> Gen. last  | 14.6      | 23.6 | 8.3   | 25.1   | 20.8  | -    | 14.3  | 0.66 |
> | Und. Only                | 34.0      | 41.0 | 37.4  | 68.9   | 68.9  | -    | -     | -    |
> | GenLLaVA                 | 30.5      | 37.1 | 35.8  | 64.5   | 66.8  | 66.9 | 12.5  | 0.73 |
>
> **Table D**: Different recipes for training. We assume that the regular alignment stage is done first using LLaVA-Pretrain-558K data except for GenLLaVA since it is single-stage recipe.
>
> ---
>
> **Performance Gaps with State-of-the-Art Models**
>
> With access to larger-scale understanding and generative datasets, we hypothesize that GenLLaVA’s performance could improve significantly. However, quantifying the exact data requirements to achieve the same results would be nothing more than speculation. See also limitations sections where we compare to other state-of-the-art models
>
> ---
>
> **Open-Sourcing Resources**
>
> We will release all resources, including datasets, code, and model checkpoints, upon acceptance. The data reuses the LLaVA-Pretrain and LLaVA-Finetune formats, with additional fields for generation tasks (e.g., `output_image`).
>
>
> ---

---

> > ### Comment · Reviewer_uJ5q · 2024-11-25
> >
> > Thank you to the authors for their efforts during the rebuttal. However, my key concerns regarding W1, W2, and Q2 remain unresolved. Specifically:
> >
> > + Q1: While I acknowledge that GenLLaVA incorporates some unique design, the modifications are not substantial enough to meet the required bar.
> > + Q2: Since the primary contribution of this work is the dataset, the lack of access to it during the review process makes it challenging to fully assess its value.
> > + W1: The authors have not provided a practical explanation or solution for how GenLLaVA would perform with stronger models.
> >
> > Therefore, I have decided to lower my score to 5.

---

### Official Review · Reviewer_swzY · 2024-11-02

**Soundness:** 1
**Presentation:** 2
**Contribution:** 2
**Rating:** 3
**Confidence:** 5

**Summary:**

GenLLaVA, a novel approach aimed at enhancing the zero-shot capabilities of a single unfied large multimodal model to perform image understanding, image generation, and image editing tasks is proposed with the following contributions:
1. Construction of a new multimodal instruction-following dataset that combines image understanding, image generation, and image editing data.
2. A unified model, GenLLaVA, is trained in a single-stage training process, which is a departure from its predecessor, LLaVA.
3. GenLLaVA outperforms LLaVA and is competitive with native multimodal models like Unified-IO 2, setting a new benchmark for building advanced general-purpose visual assistants by effectively reusing existing multimodal models.

**Strengths:**

Originality:
- New Dataset Curation: The creation of a new multimodal instruction-following dataset that amalgamates image understanding, generation, and editing is innovative. It addresses the need for diverse training data to support complex multimodal tasks.
- Single-Stage Training Strategy: Moving away from the traditional multi-stage training pipeline to a single-stage training recipe is a significant departure that simplifies the training process while maintaining performance.

Quality:
- State-of-the-Art Comparisons: With new dataset, GenLLaVA establishes its competitive standing within the field comparing to LLaVA in multi-modal understanding, GILL, AnyGPT, Unified-IO

Significance:
- The work moves towards the development of a unified MLLM, which is able to simultaneously perform image generation and image understanding. This is a rather important direction for the development of MLLM.

**Weaknesses:**

1. Conciseness of the Methodology Section: The methodology section lacks sufficient depth, particularly in explaining how the model integrates outputs from large language models with Diffuser models for tasks in image generation and editing. Specific details, such as the configuration of attention masks, inputs and target outputs during image generation, and the loss functions employed, are absent. Including these would enhance clarity.

2. Limited Originality in Methodology: Due to its brevity, the methodology section does not adequately establish the originality of the proposed approach. A more detailed explanation of the unique methods used is essential to distinguish this work from similar studies, such as GILL[1] and Unified-IO 2[2] As depicted in Figure 1, the only observable difference between GLLaVA and GILL appears to be the choice of image encoders, language models, and Diffuser models—none of which are novel. If these component substitutions introduce significant innovation, or if additional design elements are integral, these should be clearly outlined to affirm the model’s originality in achieving image understanding, generation, and editing capabilities.

3. Experimental Support for Claims: The experiments presented do not convincingly substantiate the paper's claims of achieving image understanding, generation, and editing without compromising individual performance. As Table 1 indicates, the model underperforms in image generation tasks compared to its peers, with no comparison provided against other multimodal understanding models. Additionally, while the model uses data from ShareGPT[3], the performance is lower, scoring 66.8 on MMB[4] for a 7B LLM compared to ShareGPT’s 68.8. If this discrepancy is due to factors beyond inter-task conflicts, ablation studies could help clarify. Further, the single-stage pipeline is chosen and no ablation is performed, the author follows Prismatic vlms[5]. It's a main contribution claimed in paper which should contains some experiments to support it.

4. Inadequacy of Ablation Studies: The ablation studies presented are insufficient in addressing inter-task conflicts. Table 2(a) reveals a significant drop in performance when handling all three tasks concurrently. Although the authors propose enhancing understanding performance by adding more understanding data, no specific performance comparison is provided for image understanding alone. Moreover, the use of Task Tokens boosts performance in Table 2(a), but the underlying reasons for this improvement remain unexplored. This suggests that explicit task format definition is essential during inference, warranting further analysis in the discussion.


[1] Jing Yu Koh, Daniel Fried, and Russ R Salakhutdinov. Generating images with multimodal language models. In NeurIPS, 2024.

[2] Jiasen Lu, Christopher Clark, Sangho Lee, Zichen Zhang, Savya Khosla, Ryan Marten, Derek Hoiem, and Aniruddha Kembhavi. Unified-io 2: Scaling autoregressive multimodal models with vision, language, audio, and action. In CVPR, 2024.

[3] Lin Chen, Jisong Li, Xiaoyi Dong, Pan Zhang, Conghui He, Jiaqi Wang, Feng Zhao, and Dahua Lin. Sharegpt4v: Improving large multi-modal models with better captions. In ECCV, 2024.

[4] Yuan Liu, Haodong Duan, Yuanhan Zhang, Bo Li, Songyang Zhang, Wangbo Zhao, Yike Yuan, Jiaqi Wang, Conghui He, Ziwei Liu, et al. Mmbench: Is your multi-modal model an all-around player? In ECCV, 2024.

[5] Siddharth Karamcheti, Suraj Nair, Ashwin Balakrishna, Percy Liang, Thomas Kollar, and Dorsa Sadigh. Prismatic vlms: Investigating the design space of visually-conditioned language models. In ICML, 2024.

**Questions:**

1. Could you provide a detailed explanation of how the model integrates outputs from large language models with Diffuser models specifically for image generation and editing?
2. The paper references similarities with existing models like GILL and Unified-IO 2, particularly in Figure 1, where the only discernible differences appear to be the use of certain image encoders, language models, and Diffuser models. Could you elaborate on the original aspects of GLLaVA, if any, beyond component substitution?
3. While the paper claims to achieve balanced performance across image understanding, generation, and editing tasks, Table 1 suggests that the model falls short of peers in image generation, and no comparisons are made against other multimodal understanding models. Could you address this by clarifying if factors other than inter-task conflicts influence this performance gap?
4. Your decision to adopt a single-stage pipeline, inspired by Prismatic VLMS, is an essential component of the methodology, yet no ablation studies are provided to justify this choice over a multi-stage approach. Could you explain if any experiments were conducted to compare the performance of the single-stage pipeline with a more conventional multi-stage process?
5. Table 2(a) indicates a notable performance improvement with Task Tokens, yet no explanation is provided for this boost. Could you delve into the role of Task Tokens in explicitly defining the task format during inference?
6. The paper proposes an improvement in understanding capabilities by adding more data, yet no specific comparison is provided for image understanding performance alone. Would you be able to offer more insights into the impact of this additional data specifically for image understanding?

**Details Of Ethics Concerns:**

No thics concerns.

---

> ### Author Response · Authors · 2024-11-21
> **Response to Reviewer swzY**
>
> We thank the reviewer for the feedback and comments. Below are our responses.
>
> **Conciseness of the Methodology Section**
>
> In the revised manuscript, we will:
> - Elaborate on the loss functions used for both visual understanding and generation tasks.
> - Provide details on how classifier-free guidance (CFG) is used in our model to combine outputs from the large language model and the Diffuser model.
> - Clearly outline the configuration of attention masks and the specific inputs and target outputs during the image generation and editing phases.
> ---
> **Limited Originality in Methodology**
>
> We emphasize the following to be our key contributions:
> - **Model Architecture** : Our approach is distinct from Unified-IO 2 (Encoder-Decoder) as it adopts a decoder-only architecture, reuses existing models, and does not require training from scratch, thereby significantly reducing computational costs.
> - **Dataset Contribution** : We compiled a novel fine-tuning dataset that integrates multimodal instruction tuning data for image understanding, generation, and editing. Unlike existing works, this dataset enables researchers to bypass the need for collecting and combining individual datasets manually.
> - **Diffuser Model Integration** : GILL uses a less capable LLM (OPT-6.7B) and noisier data (CC3M). Unlike GILL, which relies on text prompts to guide the diffusion model, GenLLaVA directly injects LLM features into the visual encoder (U-Net), enabling visual guidance and editing capabilities. Text-only guidance limits editing, as visual input must first be converted into text-embedings.
> ---
> **Experimental Support for Claims**
>
> Some trade-offs in understanding performance are observed. To address this:
> - We will clarify that our claim is specific to maintaining a balance across modalities, not absolute superiority in all metrics.
> - The observed performance gap (e.g., a 2% difference on MMB) is within an acceptable margin, demonstrating that the trade-offs are minor.  We will qualify our claims to make this observation.
> ---
> **Single-Stage Pipeline Justification**
>
> Our choice of a single-stage pipeline was motivated by empirical observations and challenges with multi-stage pipelines, we only cite PrismaticVLMs, as the first to do so but only for image understanding tasks, while we extend this to image generation and editing tasks.
> - Multi-stage recipes (e.g., separating multimodal alignment, generative capability addition, and instruction tuning) often resulted in catastrophic forgetting, severely degrading performance on tasks. For example, two three-stage pipelines yielded poor results, see Table A.
> ---
> **Inadequacy of Ablation Studies**
>
> We will strengthen our ablation study section by:
> - Adding a comparison of task-specific performance with and without additional understanding data. See Table A for specific results.
> - Highlighting the role of task tokens in mitigating inter-task conflicts. Task tokens explicitly define the required task during inference, reducing ambiguity and enhancing task-specific performance. This is consistent with findings from similar models like Unified-IO 2 and DreamLLM, Show-O, AnyGPT all of which use analogus strategies.
> ---
>
> **Role of Task Tokens**
>
> Task tokens serve as a simple yet effective strategy to condition the model explicitly for each task, bypassing the need for the model to infer user intent. This prevents errors in multi-modal understanding tasks as greatly present in Chameleon (no task tokens). Other strategies, such as releasing task-specific checkpoints (e.g., Emu models that release generation and understanding checkpoint separately), achieve similar goals but sacrifice the model's generality. We will expand the discussion of this mechanism in the manuscript.
>
> ---
> **Specific Performance Comparison for Image Understanding**
>
> Section 4.7 already includes comparisons of our model against state-of-the-art methods for multimodal understanding tasks. However, we will further elaborate on how the added data enhances image understanding. Specific results are detailed in Table A.
>
> | Recipe Type              | MathVista | MMMU | MMVet | SEED-B | MMB  | EVR  | CC3M | COCO |
> |--------------------------|-----------|------|-------|--------|-------|------|-------|------|
> | Gen. first -> Und. last  | 15.5      | 29.6 | 13.0  | 44.5   | 32.9  | -    | 15.9  | 0.64 |
> | Und. first -> Gen. last  | 14.6      | 23.6 | 8.3   | 25.1   | 20.8  | -    | 14.3  | 0.66 |
> | Und. Only                | 34.0      | 41.0 | 37.4  | 68.9   | 68.9  | -    | -     | -    |
> | GenLLaVA                 | 30.5      | 37.1 | 35.8  | 64.5   | 66.8  | 66.9 | 12.5  | 0.73 |
>
> **Table A**: Different recipes for training. We assume that the regular alignment stage is done first using LLaVA-Pretrain-558K data except for GenLLaVA since it is single-stage recipe.

---

### Author Response · Authors · 2024-11-28
**Main response and summary of changes in revised manuscript.**

We thank all reviewers for their feedback and comments, which have helped us improve our work. Below, we summarize the changes we have made in the revised manuscript in response to the reviewers suggestions:

- **Clarification of Claims**: We have removed unfounded claims and contextualized our contributions comparing to similar models, highlighting the novelty and significance of our approach.
- **Expanded Methodology Explanations**: We now include detailed derivations of the main loss functions and elaborate on key architectural choices, such as classifier-free guidance and task tokens.
- **Comparison Expansion**: The revised manuscript compares GenLLaVA to additional models (e.g., Emu2-34B, SEED-X, Show-o, Janus, among others) to provide a broader perspective on its performance.
- **Single-Stage Recipe Justification**: We conducted a new ablation study, demonstrating that our single-stage recipe outperforms two three-stage recipes while reducing computational complexity.
- **Comprehensive Task Token Discussion**: We expanded the discussion on task tokens, providing a more in-depth justification for their role in task-specific conditioning.
- **Performance Baselines**: We added results from additional baselines, addressing gaps noted by the reviewers.

---

### Meta-Review · Area_Chair_HrZS · 2024-12-14

**Metareview:**

This paper introduces GenLLaVA for unified image understanding and generation. After rebuttal, it received scores of 3356. Several major concerns still remain: (1) very limited novelty of the model, and (2) weak experimental results given nowadays a large volume of papers on this topic. The rebuttal is unfortunately not convincing enough. Therefore, the AC would like to recommend rejection of the paper.

**Additional Comments On Reviewer Discussion:**

All the reviewers basically share the same concerns. First, limited novelty. The proposed model is essentially the same as GILL. The novelty is very limited considering a large number of advanced models introduced this year that can achieve unified image understanding, generation, and editing. Second, weak results. This paper only compares a few outdated and underperforming models as baselines. The authors did not address well on these concerns.

---

### Decision · Program_Chairs · 2025-01-22

Reject